# Reductive and Oxidative UV Degradation of PFAS—Status, Needs and Future Perspectives

**Muhammad Umar**

Norwegian Institute for Water Research (NIVA), Økernveien 94, NO 0579 Oslo, Norway;
muhammad.umar@niva.no

**Abstract:** Perfluoroalkyl and polyfluoroalkyl substances (PFASs) consist of a group of environmentally persistent, toxic and bio-accumulative organic compounds of industrial origin that are widely present in water and wastewater. Despite restricted use due to current regulations on their use, perfluorooctanoic acid (PFOA) and perfluorooctane sulfonic acid (PFOS) remain the most commonly detected long-chain PFAS. This article reviews UV-based oxidative and reductive studies for the degradation of PFAS. Most of the UV-based processes studied at lab-scale include low pressure mercury lamps (emitting at 254 and 185 nm) with some studies using medium pressure mercury lamps (200–400 nm). A critical evaluation of the findings is made considering the degradation of PFAS, the impact of water quality conditions (pH, background ions, organics), types of oxidizing/reducing species, and source of irradiation with emphasis given to mechanisms of degradation and reaction by-products. Research gaps related to understanding of the factors influencing oxidative and reductive defluorination, impact of co-existing ions from the perspective of complexation with PFAS, and post-treatment toxicity are highlighted. The review also provides an overview of future perspectives regarding the challenges in relation to the current knowledge gaps, and future needs.

**Keywords:** PFAS; UV; VUV; oxidation; reduction; water



## 1. Introduction

Perfluoroalkyl and polyfluoroalkyl substances (PFASs) (C5–C18) are widely used in different industrial applications (clothing, paper packing, non-stick cookware, food packaging, pesticide formulations, waterproof fabrics, fume suppressants, photographic films, masking tape, firefighting foams) due to their unique properties [1,2]. These special properties of PFASs are associated with characteristics such as: (1) the hydrogen atoms on the alkyl chain are replaced by fluorine atoms [3], and (2) the presence of both long hydrophobic perfluorinated ($C_nH_{2n+1}$) carbon chain and hydrophilic functional group ($-SO_3^-$, $-COO^-$), i.e., in PFOS and PFOA [4]. PFAS have been found in both influent and effluent of wastewater treatment plants which are considered as one of the major sources for their occurrence in surface and groundwater [5–7]. Like several other micropollutants, PFAS are found at very low concentrations [8], but their refractory nature and unique physicochemical properties (Table 1) exacerbates the challenge of their degradation and/or removal.

**Table 1.** Physicochemical properties of various PFASs (adapted from Espana et al. [9]).

| Property | PFOA (Free Acid) | PFOS (Potassium Salt) |
|---|---|---|
| * Physical description | White powder/waxy white solid | White powder PFOA |
| Molecular formula | $C_8HF_{15}O_2$ | $C_8HF_{17}O_3S$ |
| Molecular weight (g mol$^{-1}$) | 414 | 538 |
| Water solubility at 25 °C (mg L$^{-1}$) | $9.5 \times 10^3$ | 680 |
| Melting Point (°C) | 45–50 | >400 |
| Boiling point (°C) | 189–192 | Not measurable |
| Vapour pressure at 25 °C (Pa) | 4.2 | $2.48 \times 10^{-8}$ |
| Organic–carbon partition coefficient (log $K_{oc}$) | 2.06 | ** 2.57 |
| Henry's law constant (atm-m$^3$ mol$^{-1}$) | Not measurable | $3.05 \times 10^{-9}$ |
| Half-Life | *** 90 days, **** >92 years (at 25 °C) | *** 114 days, **** >41 years (at 25 °C) |

* At room temperature and atmospheric pressure, ** Value estimated based on anion and not the salt, *** Atmospheric, **** In water.

An increased focus has been made on the need for adopting regulations for the release and therefore treatment of PFASs. In the conference of the parties of the Basel, Rotterdam and Stockholm Convections, PFOA and PFOS were added and amended, respectively, to the list of Convention on Persistent Organic Pollutants [10]. However, despite limited use of the PFAS compounds, their use continues in "metal plating, firefighting foams, insect baits, photographic coating, gas filter and medical devices" [10]. In 2016, US EPA issued Lifetime Health Advisories of 70 ng L$^{-1}$ for combined concentrations of PFOA and PFOS in drinking water [11]. Recently, U.S. EPA formulated PFAS Action Plan for addressing regulatory uncertainty and moving towards identifying the Maximum Contaminant Level for PFOA and PFOS [12].

Introduction of new regulatory guidelines requires that the costs of meeting new standards are justified and this together with inefficiency of most conventional treatments in degrading/removing PFAS has led to extensive research on developing appropriate remedial technologies. Several treatment approaches have been tested for the degradation and removal of PFASs including adsorption (activated carbon, carbon nanotubes), ion exchange resin, filtration (RO and NF), electrochemical oxidation, sonolysis and chemical oxidation and reduction [13]. This review focuses on the UV-based advanced oxidation and reduction process for the degradation of PFAS. While conventional UV-based processes that are based on the generation of $^\bullet$OH are known be ineffective in directly oxidizing PFAS, alternative UV-based oxidative and reductive processes have been extensively investigated for their degradation. This review looks at the different approaches that have been investigated for improving the homogeneous UV-based processes including oxidative and reductive technologies for degrading PFASs.

Several review papers have been published reporting different approaches for the removal and degradation of PFASs using different processes [9,14–17]. However, most of the reviews have been very wider in scope and thus limited emphasis has been placed on the fundamentals of oxidative and reductive processes for PFAS degradation. This paper specifically reviews these two approaches in relation to their potential and limitations for PFAS degradation. Since direct photolysis and direct $^\bullet$OH radical attack during commonly used oxidants such as $H_2O_2$ is inefficient in degrading PFAS, other oxidants such as KI, persulfate, ferric ions and carbonates are investigated by several authors that are reviewed with particular focus given to mechanism of PFAS degradation. The impact of different water quality parameters and experimental conditions (pH, reaction atmosphere, concentration of PFAS, lamp power, lamp type) on degradation kinetics is discussed. Additionally, the formation and evolution of reaction by-products under different experimental conditions is reviewed. The challenges that degradation of PFASs poses using UV-based oxidative and reductive processes are highlighted. Finally, research gaps are identified, and future recommendations are made for enhanced overall removal and/or degradation of PFAS.

## 2. Photolysis and Photochemical Decomposition Using UVC/VUV

### 2.1. Direct Photolysis (UV/VUV)

UV irradiation is divided into four regions: UVA (315–400 nm), UVB (280–315 nm), UVC (200–280 nm) and vacuum UV (VUV, 100–200 nm) [18]. Low pressure (LP) mercury lamps emitting UVC irradiation primarily at 254 nm have been investigated for the degradation of various contaminants including PFAS. While it is an efficient process for a range of compounds, conventional direct photolysis is known to be inefficient for the degradation PFAS as demonstrated by several investigations [19–22]. Little to no degradation during direct UV photolysis is attributed to insufficient breakdown of the C-F bond by photo energy generated during UV irradiation [23]. Giri et al. [23] corroborated these results and found negligible degradation of PFOA after 5 h of irradiation at 254 nm. The absorption of UV light at wavelength higher than 220 nm is very low resulting in direct photolysis being ineffective for PFAS degradation [24]. Similarly, Chen et al. [20] observed that the UV absorbance of PFOA from 190 to 280 nm (UVC) and reported a much stronger UV absorption in the VUV region below 200 nm.

Giri et al. [23] calculated the photon energy for both UVC (471.1 kJ mol$^{-1}$) and VUV (185 nm) (646.8 kJ mol$^{-1}$) and considering the C-C bond energy (347 kJ mol$^{-1}$), PFOA is prone to breakage by both wavelengths. However, C-F bond with the higher bond energy (552 kJ mol$^{-1}$) is unlikely to be cleaved by 254 nm. It is therefore established that direct UV photolysis at 254 nm is not a suitable process for the degradation of these compounds. Considering the higher photon energy generated during VUV photolysis, direct photolysis by VUV is more promising. For example, PFOA was reported to be effectively photolyzed by VUV in several findings [19–21]. VUV irradiation leads to cleavage of the C-C bond followed by the formation of fluoride ion and short-chain perfluoroalkyl carboxylates (Equations below) [20].

$$C_7F_{15}COOH \text{ (PFOA)} + hv \text{ (185 nm)} \rightarrow C_7F_{15}{}^\bullet + {}^\bullet COOH \tag{1}$$

$$C_7F_{15}{}^\bullet + H_2O \rightarrow C_6F_{13}COOH \text{ (PFHpA)} + F^- \tag{2}$$

$$C_6F_{13}COOH + hv \rightarrow C_6F_{13}{}^\bullet + {}^\bullet COOH \tag{3}$$

$$C_6F_{13}{}^\bullet + H_2O \rightarrow C_5F_{11}COOH \text{ (PFHxA)} + F^- \tag{4}$$

According to Chen et al. [20], the degradation of PFOA (25 mg L$^{-1}$) upon VUV irradiation at 185 nm (15 W) was about 62% with 17% defluorination (conversion of fluorine to fluoride) after 2 h whereas 254 nm led to negligible degradation under similar conditions. PFOA (initial concentration of 1 mg L$^{-1}$) degradation reported by Giri et al. [23] using 20 W lamp was higher (79%) compared with Chen et al. [20]. The defluorination ratio was however fairly similar (17–18%). Much lower defluorination ratio compared with the degradation of parent compound indicates that only a small proportion of PFOA was mineralized. Considering a large difference in the initial concentration of PFOA in these two studies, the effectiveness of VUV process was not significantly different considering comparable defluorination efficiency despite some difference (17%) in the PFOA degradation.

Although there are limited studies using VUV irradiation, the findings suggest that the process has the potential to degrade PFAS. Considering stronger UV absorption below 200 nm, the application of VUV photolysis requires to be investigated further using specially designed system emitting most irradiation at 185 nm such as in the case of Giri et al. [23] who reported 97% VUV emission when used a synthetic fused silica glass tube. It is worth noting that the VUV process generates ozone that could also lead to the degradation of PFAS. However, none of the studies have looked at the generation and potential degradation of PFAS related to ozone produced during VUV process. Furthermore, it has been previously demonstrated that the generation of H$_2$O$_2$ is enhanced during UV/VUV process in the presence of oxygen but was negatively impacted under alkaline conditions and in the presence of anions [25]. The addition of chemicals could therefore be

avoided during VUV process due to in-situ generation $H_2O_2$ [26,27]. Hence, it is important that VUV process is investigated with a particular focus given to optimizing conditions for greater generation of $H_2O_2$. The role of ozone generated during VUV process also needs to be investigated. Additionally, photodegradation of PFOS and PFOA could be improved by the addition of an oxidant using UV and VUV irradiation as discussed in the following section.

## 2.2. Photochemical Oxidation Using UVC/VUV

### 2.2.1. UV/$H_2O_2$

One of the most commonly used UV-based advanced oxidation processes (AOPs) that has been investigated for the degradation of recalcitrant organic contaminants is UV/$H_2O_2$. However, the addition of $H_2O_2$ is found to be unfavorable for the degradation of PFOA due to competitive absorption of photons [28]. The upper limit for the second-order rate constant of $^{\bullet}$OH reaction with PFOA ($k_{\bullet OH}$ + PFOA) is $\leq 10^5$ L·mol$^{-1}$·s$^{-1}$, which is several orders of magnitude slower than the $^{\bullet}$OH reaction with most hydrocarbons [29]. The reaction of $^{\bullet}$OH with organic compounds principally involves three mechanisms: (1) hydrogen abstraction yielding carbon-centered radicals, (2) electrophilic addition of $^{\bullet}$OH to unsaturated carbon-carbon bonds, and (3) electron transfer in which case the $^{\bullet}$OH receives an electron from the organic substituent [30]. Reaction of $^{\bullet}$OH with saturated organic compounds occurs through hydrogen abstraction and unsaturated organic compounds via radical addition. Since PFOA and PFOS has no hydrogen for abstraction, $^{\bullet}$OH can only react via direct electron transfer pathway leading to formation of much less thermodynamically favored HO$^-$ ($E^0$ = 1.9 V). The perfluorination or substitution of organic hydrogen atoms for fluorines make the PFASs inert to $^{\bullet}$OH oxidation [31]. Convectional UV/$H_2O_2$ process is therefore not considered efficient for the degradation of PFOA and PFOS. This has prompted research into using other oxidative agents in combination with UV irradiation. A summary of oxidative studies for the degradation of different PFAS compounds is given in Table 2.

For example, Thi et al. [22] investigated the degradation of PFOA by using 254 nm UV in the presence of $CO_3^{\bullet-}$ and compared the decomposition and defluorination efficiency under different pH values, initial concentration of PFOA and reaction time. Direct photolysis without $CO_3^{\bullet-}$ gave ~52% PFOA degradation after 12 h whereas when UV was used in combination with $H_2O_2$, a lower degradation (~32%) was achieved. It is known that the photolysis of PFOA can generate electron which reacts with $H_2O_2$ and $^{\bullet}$OH leading to reduced efficiency of PFOA degradation [28]. Addition of $CO_3^{\bullet-}$ to UV/$H_2O_2$ process resulted in 100% decomposition. Similarly, the defluorination efficiency was the highest (~82%) after UV/$H_2O_2$/$CO_3^{\bullet-}$, followed by UV alone (38.3%) and UV/$H_2O_2$ (27.9%). These findings indicate that $CO_3^{\bullet-}$ could be an efficient oxidant when used in the UV/$H_2O_2$ process.

The effect of pH during UV/$H_2O_2$ process in the presence of $CO_3^{\bullet-}$ was also investigated [22] with findings suggesting little difference in the degradation efficiency at pH 4.09 (97%) and 8.8 (100%) compared with pH 11 (82.4%). Likewise, the defluorination efficiencies were 72%, 83.2% and 65% at pH 4.09, 8.8 and 11, respectively. The pseudo-first-order rate constant was 0.37 h$^{-1}$ at pH 8.8 which was significantly higher than at pH 4.09 (0.27 h$^{-1}$) and pH 11 (0.076 h$^{-1}$). These findings demonstrate the effectiveness of $CO_3^{\bullet-}$ in degrading PFCAs under investigated conditions but its efficiency in real water and wastewater matrices needs to be investigated.

**Table 2.** Photooxidative studies for the degradation of PFAS under studied conditions.

| Com pound | Concentra tions (uM) | Matrix | pH | UV Source | Wavelen gth (nm) | Oxidative Agent | Degrda tion (%) | Treatment Time (h) | Lamp Power (W) | Reference |
|---|---|---|---|---|---|---|---|---|---|---|
| PFOA | 36 | Ultrapure water | 3 to 4 | VUV | 185, 254 | $Fe^{3+}$ | 100 | 72–144 | 5 | [32] |
| PFOA | 150 | Ultrapure water, surface water, WW | 2.8 | UV | 254 | $SO_4^{\bullet-}$ | 85.6 | 8 | | [33] |
| PFOA | 36 | milliQ | 3 to 4 | VUV | 185, 254 | $Fe^{3+}$ | 51.2 | 4 | 12 | [34] |
| PFOS | 20 | milliQ | 3.6 | LP | 254 | $Fe^{3+}$ | ~100 | 48 | 23 | [35] |
| PFOA | 48 | milliQ | 2 | LP | 254 | $Fe^{3+}$ | 100 | 8 | 14 | [36] |
| PFOA | 20 | DI | 5 | LP | 254 | $Fe^{3+}$, $S_2O_8^{2-}$ | 93.9 | 5 | 9 | [37] |
| PFOA | 120 | milliQ | 4.09, 8.8, 11 | LP | 254 | $CO_3^{\bullet-}$ | 100 | 12 | 400 | [22] |
| PFOS | 200 | Ultrapure water | 3.0–11 | LP | 254 | $S_2O_8^{2-}$ | 23.5 | 12 | 15 | [38] |
| PFOA | 48 | milliQ | 4.6 | LP | 185 | $Fe^{3+}$ | 98 | 48 | 12 | [39] |
| PFDeA | 100 | DI | NG | LP | 185, 254 | $S_2O_8^{2-}$ | 100 | 6 | 23 | [40] |
| PFOA | 20 | DI | 3 | LP | 254 | $Fe^{3+}$ | 92.5 | 5 | 9 W | [41] |
| PFOA | 48 | DI water | 3.5 to 4 | VUV | 185, 254 | $Fe^{3+}$ | 78.9 | 4 | 23 W | [42] |
| PFCA | 67.3 | milliQ | 1.5 | MP | 220–460 | $Fe^{3+}$ | 71.2 | 24 | 200 W | [24] |
| PFOS | 40 | milliQ | | LP | 254 | alkaline 2-propanol | 92 | 10 d | 32 W | [43] |
| PFOA/PFNA | 29.6 | Aqueous solu-tion/wax sample | 3.0–3.1 | UV | 220–460 | $S_2O_8^{2-}$ | 100 | 4 | 200 W | [44] |

In addition to investigating the degradation of conventional PFASs, some studies have looked at the degradation of compounds that are used as alternatives to PFAS. For example, the degradation of 2-(1,1,2-trifluoro-2-hepta fluoropyloxy-ethylsulfonyl)-ethanol (TFHFESE), a fluorotelomer alcohol, was recently investigated using UV photolysis and UV/$H_2O_2$ processes [45]. TFHFESE is used in textile, leather and paper production. The authors used two different UV intensities, i.e., 2.1 and 3 mW cm$^{-2}$ and reported a fairly similar level of TFHFESE degradation in DI water after 2 h, i.e., 93% and 97.8%, respectively. The degradation was enhanced when UV (3 mW cm$^{-2}$) was combined with $H_2O_2$ (25 mM), yielding 97.2% degradation after 45 min of irradiation which was 23.5% more than UV photolysis alone under similar conditions. Indirect photolysis was hypothesized to be the main mechanism of degradation during UV/$H_2O_2$ processes which was mainly driven by the generation of $^{\bullet}$OH, perhydroxyl radicals (HO$_2^{\bullet}$) and superoxide anion (O$_2^{\bullet}$). The authors also conducted preliminary assessment of the impact of water matrix using river water. While the degradation of TFHFESE was lower at initial stages of reaction (30 min), the degradation levels were similar (~98%) in DI and river water after 2 h during direct UV photolysis. Considering the river water contained considerable concentration of organics (TOC, 14.8 mg L$^{-1}$), these results are encouraging but warrant further investigation since using prolonged UV treatment is not economically feasible.

Although UV/$H_2O_2$ is generally not considered efficient for the degradation of PFAS, the process could be effective under scenarios that are favorable, for example when water contains $CO_3^{\bullet-}$. However, the fact that real water matrix contains several constituents in addition to $CO_3^{\bullet-}$, the efficiency of the process might be negatively impacted depending on the concentration of $CO_3^{\bullet-}$ and co-existing constituents. Further research is therefore needed

to understand the complexity of the interactions of different water quality characteristics on the $UV/H_2O_2$ in the presence of $CO_3^{\bullet-}$. PFAS degradation could also be significantly enhanced by combining UV with other processes as investigated by Li et al. [14] in which the authors combined photocatalysis with ozone and electrocatalysis. Since these processes are outside the scope of this review article, the authors are encouraged to refer to the mentioned study for process details and degradation mechanisms.

### 2.2.2. UV/VUV/Sulfite

UV and VUV have been investigated for PFAS degradation using sulphate radicals ($SO_4^{\bullet-}$). With electron reduction potential of 2.3 V, it reacts via direct electron transfer to generate sulphate radical anions [46]. These radicals with a quantum efficiency of unity are strong oxidants [47].

$$S_2O_8^{2-} + hv \ (< 270 \text{ nm}) \rightarrow 2SO_4^{\bullet-} \tag{5}$$

$$SO_4^{\bullet-} + e^- \rightarrow SO_4^{2-} \left( E^0 = 2.3 \text{ V} \right) \tag{6}$$

These radicals have been investigated for the degradation of both PFOA and PFOS using low and medium pressure mercury UV lamps. Chen and Zhang [19] investigated the degradation of PFOA using persulfate ($S_2O_8^{2-}$) by 185 nm and 254 nm ($K_2S_2O_8 = 407$ mg $L^{-1}$, temperature = 25 °C) achieving 84% and 65% of PFOA degradation, respectively. VUV irradiation (185 nm) was considered to achieve degradation by both direct photolysis and $SO_4^{\bullet-}$ oxidation whereas the degradation by UV (254 nm) was attributed to $SO_4^{\bullet-}$ generated upon activation of $S_2O_8^{2-}$ under the UV irradiation. The degradation of PFOAs using a UV/visible light generated by a xenon/mercury lamp in the presence of $S_2O_8^{2-}$ was investigated by Hori et al. [44]. The decomposition rate of PFOA was 11-fold faster than direct photolysis using 50 mM $S_2O_8^{2-}$ (Hori et al. 2005). The authors found that the short-chain PFCAs formed were oxidized by $SO_4^{\bullet-}$ to $CO_2$ and fluoride ($F^-$). A complete degradation of PFOA was noted after 4 h (pH 3.0–3.1), however, the generation of $F^-$ and $CO_2$ continued such that the $F^-$ yield reached 73.8% after 12 h. After 2 h, total fluorine recovery (molar ratio of total fluorine content in $F^-$ and short-chain PFCAs as well as in unchanged PFOA to the PFOA prior to irradiation) was 99.1%. Total carbon recovery (molar ratio of total carbon in $CO_2$, generated short-chain PFCAs and in unchanged PFOA to that in the PFOA prior to irradiation) was also high (97.7%); the initial fluorine and carbon in PFOA was almost completely tracked. The effect of $S_2O_8^{2-}$ concentration was found to be linear and increasing PFOA degradation was observed with increasing $S_2O_8^{2-}$ concentration up to 0.59 mM and further increase in its concentration did not lead to any improvement in PFOA degradation. This was attributed to the saturation of $SO_4^{\bullet-}$ at higher concentrations of $S_2O_8^{2-}$.

The authors [44] also investigated the degradation of perfluorononanoic acid (PFNA) in wax solution. After 12 h, about 94% of PFNA (initial concentration 1.51 mg/L) was degraded in the presence of 50 mM $S_2O_8^{2-}$. Although it took longer to achieve this level of degradation, the effectiveness of the process was demonstrated. The first bond to be cleaved by $SO_4^{\bullet-}$ was C-C leading to the formation of $C_7F_{15}^{\bullet}$. These radicals form $C_7F_{15}OH$ due to their reaction with water which then undergoes hydrogen fluoride (HF) elimination to form $C_6F_{13}COF$ [40]. Hydrolysis of acid fluoride results in the formation of PFCAs without $CF_2$ unit, producing one $CO_2$ molecule and two fluoride ions.

A comparison between VUV alone and in combination with $K_2S_2O_8$ and $Na_2S$ under oxygen and nitrogen atmosphere, respectively, was made for the degradation of perfluorodecanoic acid (PFDeA) [40]. The degradation of PFDeA was ~60% after 5 h irradiation (light intensity of 62–69 mW cm$^{-2}$) but a much lower defluorination ratio of ~16% was achieved using VUV alone. However, in the presence of $K_2S_2O_8$ under oxygen atmosphere, the concentration of PFDeA was below the detection limit (not given) with 36% defluorination after 5 h. Increasing the $K_2S_2O_8$ concentration from 0.1 mM to 5 mM did not improve the PFDeA degradation and the formation of $F^-$ that ranged between 0.41 and 0.43 mg·L$^{-1}$·min$^{-1}$. The trend of PFDeA degradation in the presence of 0.1–5 mM $Na_2S$ un-

der nitrogen atmosphere was fairly similar to VUV/$K_2S_2O_8$ system and the concentration of PFDeA was below the detection limit for both 0.1 and 0.5 mM $Na_2S$. Both direct photolysis of PFDeA by VUV and photochemical decomposition by $SO_4^{\bullet-}$ was proposed during VUV/$K_2S_2O_8$ process with $SO_4^{\bullet-}$ playing a significant role in the oxidative degradation of PFDeA. During VUV/$Na_2S$ treatment, however, both oxidative and reductive (by $e^-_{aq}$) degradation occurred but no conclusion regarding which mechanism was most prevalent was made requiring this system to be investigated further. The PFOA degradation pathway during UV-based AOPs including $SO_4^{\bullet-}$ is given in Figure 1 [17]. Briefly, PFOA degraded to fluoride ions, formic acid and $CO_2$ via sequential removal of $CF_2$ during UV-based AOPs.

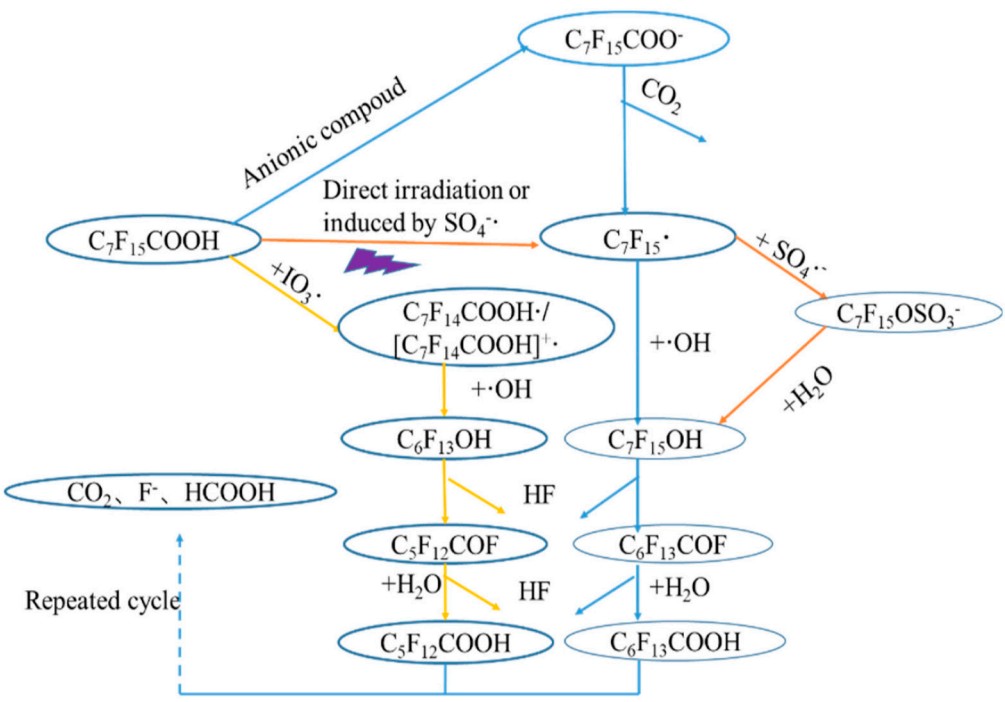

**Figure 1.** Degradation pathways of PFOA during UV-based AOPs (adapted from Wang et al. [17]).

A comparison of different treatment approaches including UV/$K_2S_2O_8$ and $Fe^{2+}$/$K_2S_2O_8$ was made by Yang et al. [38] for the decomposition of PFOS. The first-order rate constants calculated for $F^-$ generation for UV/$K_2S_2O_8$ and $Fe^{2+}$/$K_2S_2O_8$ were 0.016 and 0.010 $h^{-1}$, respectively. Hence UV/$K_2S_2O_8$ performed marginally better than $Fe^{2+}$/$K_2S_2O_8$. Since the activation of $K_2S_2O_8$ generates $SO_4^{\bullet-}$ from $S_2O_8^{2-}$, the authors determined the first-order rate constants for $SO_4^{\bullet-}$. Similar to the $F^-$ generation kinetics, the first-order rate constant for UV/$K_2S_2O_8$ was slightly greater (0.162 $h^{-1}$) than $Fe^{2+}$/$K_2S_2O_8$ (0.131 $h^{-1}$). Despite fast conversion of $SO_4^{\bullet-}$ (~13 mM after 2 h) from $S_2O_8^{2-}$ during $Fe^{2+}$/$K_2S_2O_8$, the defluorination ratio was only 23.5% after 12 h. Lower defluorination was hypothesized to be due to inadequate concentration of $SO_4^{\bullet-}$. The degradation efficiency for UV/$K_2S_2O_8$ process was not given by the authors. Increasing treatment time led to greater generation of $SO_4^{\bullet-}$ and a decrease in pH resulting in a greater total PFOS degradation. Furthermore, acidic conditions were reported to be favorable for $SO_4^{\bullet-}$ formation since it reacts with $HO^-$ to generate $^{\bullet}OH$ under basic conditions [48] which are incapable of oxidizing PFOA. The effect of the concentration of $S_2O_8^{2-}$ (0–12.5 g $L^{-1}$) was also investigated and the generation of $F^-$ was found to increase with increasing concentration of $K_2S_2O_8$ up to 10 g $L^{-1}$. During $Fe^{2+}$/$K_2S_2O_8$, the generation of $F^-$ kept increasing with increasing concertation of $K_2S_2O_8$. It was attributed to the saturation of $S_2O_8^{2-}$ achieved for UV/$K_2S_2O_8$ but not for $Fe^{2+}$/$K_2S_2O_8$. Since $SO_4^{\bullet-}$ could react with $S_2O_8^{2-}$ and $H_2O$ in addition to self-scavenging [44], the concentration of $K_2S_2O_8$ needs to be optimized depending on the type of treatment used.

Qian et al. [33] also investigated and modelled the degradation of PFOA using 254 nm UV in the presence of $SO_4^{\bullet-}$ and the formation of chlorite ($ClO_3^-$) in the presence of chloride

ion ($Cl^-$). The degradation of PFOA was 85.6% after 8 h with mineralization ranging from 35–48% (based on measurement of PFOA species and total organic carbon (TOC)) with 59% of fluoride converting to fluoride ion. Demonstrating the impact of water matrix, the authors showed that 50% degradation occurred in pure water after 2.4 h whereas similar level of degradation in wastewater matrix needed 5.5 h. In addition to the impact of water matrix, longer irradiation time needed for the degradation was also due to lower UV lamp intensity. In agreement with other studies, the degradation was shown to be stepwise process considering the evolution of intermediates; perfluoroheptanoic acid (PFHpA) and perfluoroheptanic acid (PFHeA) increased for the first 4 h with a slight decrease afterwards whereas the other intermediates (perfluoropentanoic acid (PFPeA), perfluorobutanoic acid (PFBA), PFPrA (pentafluoropropionic acid)) showed increasing concentration with increasing time of $UV/S_2O_8^{2-}$ (PS) treatment. They found that the process was efficient but was severely impacted in the presence of $Cl^-$ and the degradation was delayed until the conversion of nearly all $Cl^-$ to $ClO_3^-$. $SO_4^{\bullet-}$ preferentially reacts with $Cl^-$ due to high rate constant of $4.7 \times 10^8$ $M^{-1}s^{-1}$ [49] which was three orders of magnitude greater than the rate constant between PFOA and $SO_4^{\bullet-}$ [33]. These findings corroborate the findings of Yuan et al. [49] that the reaction between $SO_4^{\bullet-}$ and $Cl^-$ leads to reduction of $SO_4^{\bullet-}$. Likewise, $HCO_3^-$ was also found to negatively impact the PFOA degradation due to competition for reaction with $SO_4^{\bullet-}$. Although PFOA could be degraded by $CO_3^{\bullet-}$, its second-order rate constant is much lower than the reaction between $SO_4^{\bullet-}$ and PFOA [22]. However, the impact of $HCO_3^-$ was found to be much less prominent compared with $Cl^-$. The authors also investigated $UV/H_2O_2$ for comparison of the effect of $SO_4^{\bullet-}$ and $^\bullet OH$ [33]. No degradation of PFOA occurred in the $UV/H_2O_2$ process even in the presence of 15 mM $H_2O_2$.

With most studies focusing on the impact of $Cl^-$ on the $UV/SO_4^{\bullet-}$ process [50,51], the conversion of $Cl^-$ based radicals to $ClO_3^-$ has rarely been investigated. An earlier investigation looked at the formation of $ClO_3^-$ at different pH values with acidic pH favouring the formation of $ClO_3^-$ [52]. The decomposition of PS under UV irradiation generates protons that promotes the formation of $ClO_3^-$ [53] resulting in pH drop as was noticed with increasing the concentration of PS from 5 to 30 mM [33].

It is evident from the literature that the treatment time needed to achieve adequate degradation of PFAS is quite long. The degradation process could potentially be intensified by using suitable catalysts. Overall, future investigations focused on understanding the impact of competitive reactions, i.e., with organics/inorganics, and optimization of other process parameters (pH, PFAS concentration, UV intensity, dose of sulfite etc.) are needed to better understand PFAS degradation mechanisms and evolution of by-products.

### 2.2.3. UV/VUV/$Fe^{3+}$

The use of ferric ions ($Fe^{3+}$) during photodegradation of PFAS has been increasingly focused over the last few years. The radicals and complexes formed during the UV/VUV-based photo-processes have been termed as efficient in degrading PFAS. Under specific experimental conditions, as discussed later in this section, regeneration of $Fe^{3+}$ have been reported leading to enhanced PFAS degradation.

A number of studies have looked at the efficiency of $UV/Fe^{3+}$ in lab-scale investigation under different experimental condition. A detailed investigation combining UV with $Fe^{3+}$ was carried out for the degradation of PFOA at initial concentration of 48 μM [36]. The authors looked at the impact of three different initial pH values, i.e., 2, 3.7 and 5 during $UV/Fe^{3+}$ process employing a 254 nm UV lamp. The pseudo-first-order constants after 4 h was the greatest for the lowest pH of 2 (0.123 $h^{-1}$) followed by pH 3.7 (0.104 $h^{-1}$) and pH 5 (0.015 $h^{-1}$). A large decrease in pH was noted, i.e., from initial pH of 5 to 3.9 after 48 h due to the formation of formic acid in agreement with a previous study [42]. At initial pH of 3.7, the decrease in pH was relatively small with final pH value of 3.6 whereas negligible change in pH occurred for solution with initial pH value of 2. However, a significant change in the reaction rate constants occurred at all pH values with

a 30% decrease at pH 2 (0.086 $h^{-1}$) and 69% decrease at pH 3.7 (0.032 $h^{-1}$). In contrast, the reaction rate increased to 0.052 $h^{-1}$ (an increase of 29%) at an initial pH of 5 which was associated with a large decrease in pH as mentioned earlier. Complete defluorination was not achieved during first 4 h even at pH 2. However, after 8 h, >100% defluorination was achieved which was attributed to decomposition of accumulated intermediates. The defluorination ratios at initial pH of 3.7 and 5 were similar after 8 h (~30%) with a marginal improvement occurring with increasing irradiation time. Lower defluorination ratios were also reported with increasing pH by the same authors in their earlier study [39]. Ohno et al. [39] reported a strong correlation ($p < 0.01$) between $^{\bullet}OH$ generation and $Fe^{3+}$ concentration ($r = 0.989$) for the decomposition of PFOA. The formation of complex between $Fe^{3+}$ and PFOA reduced iron to $Fe^{2+}$ which in the presence of $^{\bullet}OH$ was oxidized back to $Fe^{3+}$. The regeneration of $Fe^{3+}$ was therefore related to the enhanced defluorination efficiency at pH $\leq 3.5$ as indicated by the reduced defluorination after 4 h, i.e., when no generation of $^{\bullet}OH$ occurred indicating the degradation of PFOA was solely due to $Fe^{3+}$.

The authors [36] also looked at the types of iron species present at different pH values and found that dissolved $Fe^{3+}$ was the predominant species at pH 2 prior to irradiation. At pH 3.7, only 29% $Fe^{3+}$ was in dissolved form whereas almost no dissolved $Fe^{3+}$ was present at pH 5 with $Fe(OH)^{2+}$ being the main dissolved species at these pH values. Upon irradiation, the concentration of dissolved $Fe^{3+}$ decreased whereas that of $Fe^{2+}$ decreased at initial pH values of 2 and 3.7. For pH 5, an increase in the concentration of $Fe^{3+}$ started to occur after 12 h, i.e., when the pH started to decrease. It can therefore be concluded that PFOA decomposition rates were driven by the concentration of dissolved iron species leading to the formation of PFOA-Fe complexes. These conditions were hypothesized to be present immediately for pH 2 and 3.7 as indicated by their comparable PFOA decomposition in initial stage of treatment, i.e., 4 h.

Investigating PFOA degradation (initial concentration of 20 μM) using 254 nm UV (9 W), Tang et al. [41] found that the degradation took place in two stages during $UV/Fe^{2+}/H_2O_2$, $UV/Fe^{3+}/H_2O_2$, $UV/Fe^{3+}$ and $UV/Fe^{2+}$ processes. They attributed first stage decomposition to $^{\bullet}OH$ during first 60 min of treatment characterized by complete consumption of $H_2O_2$ (30 mM) giving a degradation and defluorination ratio of 87.8% and 35.8%, respectively. The second stage involved degradation due to the generation of $Fe^{3+}$ resulting in increased degradation and defluorination ratio to 95% and 53.2%, respectively, after 5 h. Control experiments were performed using either $UV/H_2O_2$ or $Fe^{2+}/H_2O_2$ in which no fluoride ions were detected in either system after 24 h indicating no degradation of the target compound. Defluorination efficiency after $UV/Fe^{2+}/H_2O_2$, $UV/Fe^{3+}/H_2O_2$, $UV/Fe^{3+}$ and $UV/Fe^{2+}$ was 46%, 26%, 34% and 17%, respectively, after 24 h using 40 mM $H_2O_2$, and 4 mM $Fe^{2+}$ or $Fe^{3+}$ at pH 3. The $UV/Fe^{3+}$ combination achieved greater level of defluorination than $UV/Fe^{2+}$ (due to electron transfer between $Fe^{2+}$ and PFOA [42]) and $UV/Fe^{3+}/H_2O_2$ (due to consumption of $Fe^{3+}$ impeding the transfer of electrons between $Fe^{3+}$ and $H_2O_2$). Hence, the degradation of PFOA was attributed to two main mechanisms, i.e., through oxidative damage by $^{\bullet}OH$ and due to electron transfer between PFOA and $Fe^{3+}$. The degradation pathway in the presence of $Fe^{3+}$ is shown in Figure 2 below.

$UV/Fe^{2+}/H_2O_2$ being the most efficient system, the authors investigated the process further at different experimental conditions [41]. The impact of $Fe^{2+}$ (1–4 mM) and $H_2O_2$ concentration (10–40 mM) and pH (1.5–5.5) was investigated for the degradation of PFOA. Defluorination efficiency was 53.2% at optimum conditions which were 2 mM $Fe^{2+}$, 30 mM $H_2O_2$ and pH 3. The results showed that the degradation of PFOA by $^{\bullet}OH$ was not possible until the activation of PFOA by UV irradiation, i.e., $Fe^{2+}$ initiated and promoted the activation of $H_2O_2$ leading to enhanced degradation efficiency. However, generation of higher than optimum concentration of $Fe^{2+}$ could lead to scavenging of $^{\bullet}OH$ [41]. Similarly, excess $H_2O_2$ can also scavenge $^{\bullet}OH$ and it is therefore important to optimize the process to both avoid excess use of chemicals and maximize the process efficiency. The effect of pH was particularly important and pH 3 was found to be the optimum. A change in pH impacted the degradation of PFOA due to the formation of $H_3O_2^+$ and reduced

complexation between $Fe^{3+}$ and PFOA, respectively. It is worth noting that $UV/H_2O_2$ and $Fe^{2+}/H_2O_2$ led to generation of $^{\bullet}OH$ but no degradation of PFOA was observed during these processes. The authors therefore concluded that the activation of perfluorinated carboxylate anions is necessary for PFOA degradation which can be achieved by UV irradiation in the Fenton system.

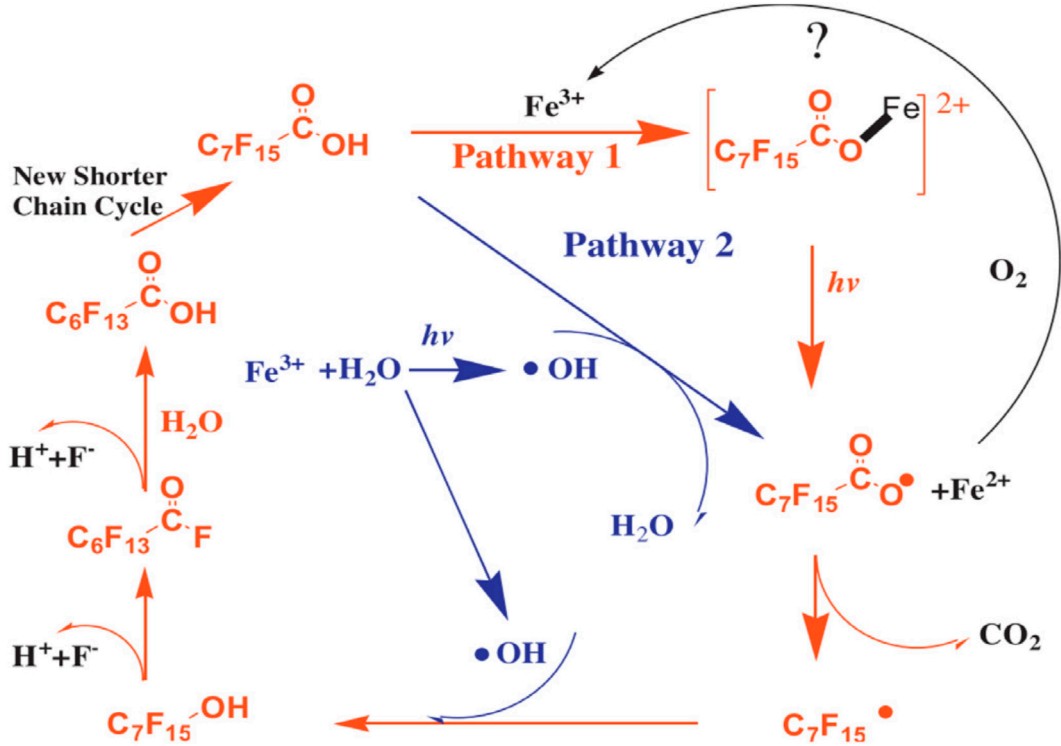

**Figure 2.** Proposed PFOA degradation pathway in the presence of $Fe^{3+}$ [54].

Another study investigated the impact of $Fe^{3+}$ (0.1 mM) on the degradation of PFOS (20 μM) [35]. The rate constant was >50-fold (1.67 $d^{-1}$) higher in the presence of $Fe^{3+}$ compared with direct photolysis (0.033 $d^{-1}$). The degradation of PFOS was only 12% after 72 h during UV photolysis whereas it decreased to below the detection limit (<1 μM) after 48 h when $Fe^{3+}$ was used. Additionally, the impact of oxygen and nitrogen was investigated on the degradation of PFOS. The impact of $H_2O_2$ was also investigated under oxygen atmosphere. It was found that both degradation of PFOS and defluorination were enhanced (absolute values were not given) due to greater generation of $^{\bullet}OH$. Describing the mechanisms of degradation, it was shown that $^{\bullet}OH$, although unable to induce PFOS degradation, could facilitate the degradation process through reaction with $^{\bullet}C_8F_{17}$ formed upon excitation of $Fe^{3+}$ and PFOS complex under UV irradiation. Although $^{\bullet}OH$ may not directly oxidize the short-chain PFAS, they could facilitate degradation by reacting with already activated PFAS [55]. For example, Huang et al. [55] found that the PFOA radicals formed upon direct electron transfer could be converted to $^{\bullet}C_7F_{15}$ that can react with $^{\bullet}OH$ to form $C_7F_{15}OH$ leading to F elimination and hydrolysis. Although a direct comparison is not possible between different studies, the findings of Jin et al. [35] and Lyu et al. [56] who used LP and MP UV lamps, respectively, it is apparent that reductive degradation is much faster than the oxidative degradation of PFOS. Similarly, a direct comparison between $UV/SO_3^{2-}$ with $UV/I^-$ for the degradation of PFOA demonstrated that the reductive process was more effective [57]. Further details are discussed in Section 3 on reductive photodegradation.

PFOA defluorination efficiency of VUV (185 nm) in the presence of $Fe^{3+}$ as catalyst was evaluated in the presence of nitrogen and oxygen, and without any gas supply [32]. VUV alone successfully achieved complete defluorination of PFOA but the time needed

was long, i.e., 144 h. However, in the presence of $Fe^{3+}$ (20 µM), total defluorination occurred in half the time (i.e., 72 h). As mentioned earlier, $Fe^{3+}$ forms complexes with PFCAs [41] resulting in photolyzing to $Fe^{2+}$ and organic radical [58]. It has been proposed that the complexation between PFOA and $Fe^{3+}$ is prerequisite for the degradation of PFOA considering the UV-Vis spectra of $Fe_2(SO_4)_3$ and PFOA [54] which agrees with the findings of Tang et al. [41]. The effect of bicarbonate ($HCO_3^-$), chloride ($Cl^-$), sulphate ($SO_4^{2-}$), nitrate ($NO_3^-$) and perchlorate ($ClO_4^-$) on the degradation of PFOA was also investigated [32] during VUV/$Fe^{3+}$ system. The impact of $HCO_3^-$ and $SO_4^{2-}$ was the greatest than all other anions with $ClO_4^-$ having negligible impact on defluorination efficiency. Considering $^\bullet OH$ as the main oxidizing specie in the system, the negative impact of $HCO_3^-$ was expected since they are strong $^\bullet OH$ scavengers. $NO_3^-$ does not scavenge $^\bullet OH$ but instead is photolyzed by VUV to generate reactive species including $^\bullet OH$ [59]. However, it is also an efficient electron receptor and therefore can lead to its preferential absorption of VUV light, thereby reducing the process efficiency [60,61]. The similar is possible for $SO_4^{2-}$ that can also absorb VUV [60]. Moreover, $Fe^{3+}$ possess strong affinity for anions that follow the order $^\bullet OH(8.95) > SO_4^{2-}(2.94) > Cl^-(0.5) > NO_3^-(-0.23) > ClO_4^-$ [62]. It is therefore expected that $SO_4^{2-}$ would negatively impact the degradation of PFOA. In fact, $SO_4^{2-}$ interferes the complexation between $Fe^{3+}$ with PFOA leading to reduced decomposition of PFOA. The presence of $Cl^-$ is also known to delay the degradation of PFOA by $SO_4^{\bullet-}$. As described earlier, $SO_4^{\bullet-}$ reacts first with $Cl^-$, and the degradation of PFOA initiates only after $Cl^-$ has changed to $ClO_3^-$ [33]. The higher the concentration of $Cl^-$, the longer it took before $SO_4^{\bullet-}$ could begin to degrade PFOA and the concentration of $SO_4^{\bullet-}$ was reported to gradually increase upon complete conversion of $Cl^-$ to $ClO_3^-$ [33]. The inhibitory effect of chloride on PFOA decomposition was attributed to the reaction between $^\bullet OH$ [29] and interference with the complexation of PFOA and $Fe^{3+}$ [39]. These findings are critical for water matrix containing chloride particularly at higher concentrations.

Similar to Liang et al. [32], Cheng et al. [34] found that the degradation of PFOA increased (in this case 2-fold) when using 20 µM $Fe^{3+}$ during VUV (185 nm) process when compared with VUV alone. A defluorination efficiency of 51.2% of PFOA (initial concentration of 36 µM) occurred in 4 h. Similarly, the defluorination rate constant for VUV/$Fe^{3+}$ (0.00308 $min^{-1}$) was ~2.6-fold greater than VUV alone (0.00119 $min^{-1}$). The impact of PFOA concentration on its degradation in the VUV/$Fe^{3+}$ process was investigated by using 4 concentrations (12, 36, 73, 144 µM); the degradation rate decreased with increasing PFOA concentration as indicated by the decreasing defluorination rate constant of 3.08, 1.86, 1.51 and $1.11 \times 10^{-3}$ $min^{-1}$, respectively. This could be attributed to two factors: (1) photon limitations considering the similar light intensity for all PFOA concentrations, and (2) low concentration of $Fe^{3+}$ (10 µM) that could have reduced the formation of complexes. In the next step, the authors looked at the impact of $Fe^{3+}$ concentration by using a range of concentrations (5–70 µM). They found that an increase in concentration of $Fe^{3+}$ up to 20 µM enhanced defluorination ratio; from ~25.7% to 51.2% with increasing concentration from 5 to 20 µM, respectively. However, further increase in the concentration resulted in decreasing defluorination efficiency. Wang et al. [42] reported that the concentration of $Fe^{3+}$ up to 80 µM had positive effect on the degradation of PFOA. Considering fairly similar experimental conditions in terms of initial concentration of PFOA and pH (Table 2), the difference in these two studies could be attributed to the lamp power which was almost double in the case of Wang et al. [42] when compared with Cheng et al. [34] (Table 2). The trend in the reduction of decomposition of PFOA at high $Fe^{3+}$ concentrations could therefore be attributed to the photon limitation due to competition between $Fe^{3+}$ and PFOA. Since no data is available in terms of the quantum yield or UV fluence, it is rather difficult to conclude if it could only be associated with photon limiting phenomenon. It stresses the need to report this very important criterion to enable direct and valid comparison between different studies. The effect of pH was also investigated using initial pH values of 2.5, 3, 4, 5 and 7 and a decreasing trend in the degradation of PFOA was found with an increase in pH [34]. A decrease in efficiency at higher pH (>4) could be attributed to precipitation

of ferric hydroxide. Furthermore, it is known that PFOA could become protonated below pH 2.8 [63] which is not favorable for complexation with $Fe^{3+}$ nor for electronic transfer indicating the most appropriate pH range is between 3 and 4.

The impact of other water constituents particularly organic matter has received more attention recently. For example, Liang et al. [32] studied the impact of dissolved organic matter (DOM) on the degradation of PFOA. The authors used 15 mg/L humic acid for up to 6 h during $VUV/Fe^{3+}$ treatment and noted a significant decrease in the defluorination efficiency in the first 2 h due to preferential degradation of humic acid. However, the impact of humic acid diminished gradually with increasing irradiation time such that the total defluorination with and without humic acid was comparable after 6 h. Hence, the initial lower defluorination was attributed to the absorption of VUV by humic acid consuming the oxidative species ($^{\bullet}OH$, $^{1}(O_2)$, etc.) resulting in its degradation as shown by the corresponding decrease in TOC and change of color from grey/black to transparent. However, humic acid can also act as photosensitizer that can assist the photodecomposition. Hence, it could lead to generation of reactive species and at the same time act as a scavenger of oxidative species and could preferentially absorb VUV. The concentration of humic acid used in their study was higher (15 mg $L^{-1}$) than normally found in most natural water matrices indicating the time needed to achieve comparable defluorination efficiency could be minimized when low content of humic acid is present. Although the role of humic acid was found insignificant for longer duration of treatment, natural water matrix contains other organics that could be more recalcitrant to photodegradation. Nonetheless, these findings are promising for targeted degradation of PFOA. For more complex water matrices, a combination of treatments could be considered for the removal of co-existing constituents (i.e., organics) that are present at much higher concentrations for their pre-removal/degradation to enhance targeted breakdown of PFOA.

UV-Fenton has been shown to be effective in degrading PFOA with some studies looking at the degradation of PFOS. It is recommended that future studies look at UV fluence or similar standard parameter(s) instead of irradiation time that provides little information with regards to the efficiency of the process and hinders the comparative assessment between different findings. The impact of conventional water quality parameters also needs to be explored. Additionally, effectiveness of Fenton-based processes is highly pH dependent and low pH is one of the limitations for practical applications. Research has shown that photo-Fenton process could be effective at near-neutral pH in the presence of chelating agents. The findings of studies carried out for other contaminants could be a good starting point to understand the degradation mechanism and kinetics of PFAS degradation during modified photo-Fenton process.

### 2.3. Reaction by-Products during UV/VUV Photodegradation

Investigating PFOA degradation using $VUV/Fe^{3+}$, Cheng et al. [34] identified intermediate by-products containing $C_2$–$C_7$ perfluoroalkyl groups in agreement with previous investigations [20,28]. The longer chain intermediates including PFHpA ($C_7$) and perfluorohexanoic acid (PFHxA) ($C_6$) reached maximum concentration after 1.5 h and 3 h, respectively, followed by decreased concentration with increasing irradiation time. The remaining intermediates ($C_2$–$C_5$) increased throughout the 4 h irradiation period. The order of concentration followed PFPeA > PFBA > perfluoropropionic acid (PFPA) > trifluoroacetic acid (TFA) demonstrating the longer chain intermediates appeared at the start of the reaction followed by decomposition to shorter chain products. Investigating PFOA degradation using 254 nm UV during $UV/H_2O_2/Fe^{2+}$ process, Tang et al. [41] found that the degradation intermediates included the short chain perfluorocarboxylic acids containing 2, 3 4, 5 and 6 carbon atoms and fluoride ions which is in agreement with others [23,64]. Liang et al. [32] also identified perfluoronated carboxylic acids with 2–7 carbon atoms during $VUV/Fe^{3+}$ degradation of PFOA. It was further noted that the shorter chain degradation products were higher in concentration, i.e., PFPA > PFBA > PFPeA > PFHxA > PFHA [32]. Jin et al. [35] while investigating the degradation of PFOS

using UV/$Fe^{3+}$ found $C_2$–$C_8$ PFCAs in addition to sulphate and fluoride as degradation by-products. Theses intermediates are similar to that reported in earlier studies using UV/Fenton [41] and VUV/$Fe^{3+}$ [32].

A comparison of the formation of the types of reaction intermediates was made by Wang et al. [40] during direct VUV photolysis and in the presence of $K_2S_2O_8$ or $Na_2S$. Both direct photolysis and photochemical decomposition showed similar types of reaction intermediates despite direct photolysis being less effective. They identified decomposition by-products as PFNA ($C_8F_{17}COO^-$), PFOA ($C_7F_{15}COO^-$), PFHpA ($C_6F_{13}COO^-$), PFHxA ($C_5F_{11}COO^-$), PFPeA ($C_4F_9COO^-$) and PFBA ($C_3F_7COO^-$) during PFDeA defluorination and degradation using $K_2S_2O_8$ or $Na_2S$ with VUV. The concentration of PFNA and PFOA increased during the first ~2 h whereas those of PFHxA, PFPeA and PFBA kept increasing throughout the reaction period of 6 h during VUV/1 mM $K_2S_2O_8$. The formation of PFHpA showed a slightly different trend with increase in concentration up to 4 h followed by a decrease thereafter. A similar trend was observed during VUV/$Na_2S$ process.

Evolution of intermediates over much longer duration has been investigated in some studies [22,38]. The concentration of PFOS and evolution of intermediates was tracked for 12 h during UV/$K_2S_2O_8$ process [38]. During the first 4 h, the concentration of PFOA, PFHpA and PFPA reached maximum levels followed by a decrease thereafter. The concentration of other intermediates (PFHxA, PFBA, PFPrA and TFA) increased with increasing time indicating the process was ineffective in degrading these compounds under investigated conditions. Hori et al. [44] reported the formation of reaction by-products during PFOA degradation using UV/visible light generated by a xenon/mercury lamp in the presence of $S_2O_8^{2-}$. The intermediate by-products included PFHpA ($C_6F_{13}COOH$), PFHxA ($C_5F_{11}COOH$), PFPeA ($C_4F_9COOH$), heptafluorobutyric acid (HFBA, $C_3F_7COOH$) that reached maximum level after 2 h. Increasing the irradiation time resulted in the formation of even shorter-chain PFCAs ($C_2F_5COOH$, $CF_3COOH$). Thi et al. [22] also followed the concentration of intermediate by-products over 12 h during UV/$H_2O_2$/$CO_3^{\bullet-}$ and found that PFHpA appeared after 30 min of treatment followed by PFHxA and PFPeA after 1 h and FPBA and PFPrA appearing after 2 h. Both PFHpA (20.7 μM) and PFHxA (15.4 μM) showed the highest concentration among other with gradual decrease after 8 and 12 h, respectively. The other intermediates showed continuous increase in the concentration throughout 12 h.

## 2.4. Photochemical Oxidation Using MP UV—Impact of Experimental Conditions

Some studies have investigated the degradation of PFAS using medium pressure UV lamps. For example, Hori et al. [24] investigated photochemical degradation using MP UV lamp (220–460 nm) of PFCAs containing 3–5 carbon atoms (PFPrA, PFBA, or PFPeA). They reported that the absorption of UV light was much higher for deep-UV region to 220 nm but was much lower for 220–270 nm. After 24 h of direct photolysis, 24.3% of PFPeA was degraded yielding 12.1% $F^-$. The other two PFCAs (PFPrA, PFBA) showed lower but comparable degradation and $F^-$ yield of about 16% and <10%, respectively. Degradation of these short chain PFAS enhanced when 5 mM $Fe^{3+}$ was used such that the degradation of PFPeA, PFBA and PFPrA was about 2.7-, 3- and 3.8-fold greater when compared with UV alone after 24 h. Similarly, the amount of $F^-$ yield was greater although it did not follow the trend of degradation. The degradation followed pseudo-first-order kinetics with increasing rate of degradation with increasing initial concentration of PFPeA demonstrating that the complexes formed between $Fe^{3+}$ and PFPeA resulted in photo-redox reactions that led to the formation of $Fe^{2+}$ and oxidized PFPeA. The degradation of PFPeA was markedly higher in the presence of oxygen (64.5%) than argon (35.6%) and so was the conversion of $Fe^{3+}$ to $Fe^{2+}$, i.e., 93.3% and 0.70%, respectively. It was concluded that oxygen was required for re-oxidation step of $Fe^{2+}$ conversion to $Fe^{3+}$ since the presence of oxygen could increase the formation of $HO_2$ that can expedite the re-oxidation process. Using three different sources of $Fe^{3+}$, the authors found that the degradation of PFPeA was comparable for iron perchlorate (71.2%) and iron sulphate (64.5%) whereas it reduced significantly using

iron chloride (24.3%). It was considered that chloride ions could interfere the formation of complexes between $Fe^{3+}$ and PFPeA as also described by others [32,39].

While the earlier investigation by Hori et al. [24] looked at the degradation of simple water matrix, a comparison of photochemical degradation of PFOS using a medium pressure lamp (500 W) was investigated in the deionized water, phosphate buffer (PBS), lake water and effluent of a municipal WWTP [65]. The degradation of PFOS was observed to be greater than in DI water with fairly similar degradation level and trends for PBS, lake water and effluent samples for the first 3 h. Continuing the treatment for another 3 h showed the maximum degradation of PFOS for lake water whereas PBS and effluent samples exhibited lower but comparable degradation with lowest degradation occurring for DI water. The degradation of PFOS was described by pseudo-first-order decay model and the decomposition rate constants calculated for WWTP effluent, lake water and PBS were $0.1 \pm 0.02 \, h^{-1}$, $0.16 \pm 0.01 \, h^{-1}$ and $0.11 \pm 0.02 \, h^{-1}$, respectively. The DI water had the lowest rate constant with $0.036 \pm 0.003 \, h^{-1}$. These results demonstrate that some substances in water could facilitate the degradation of PFOS. It was, however, noted that the defluorination ratio was independent of reaction kinetics. For example, despite showing similar kinetics, the defluorination ratio was markedly lower for WWTP effluent than PBS. Since WWTP effluent had humics and other organics (TOC, $43 \, mg \, L^{-1}$), it is plausible to assume that the defluorination inhibited despite comparable degradation of PFOA in different water matrices. The authors found that the decomposition of PFOA was negatively impacted by the presence of humic acid and low ionic strength whereas phenol and ammonia positively impacted the degradation process.

## 3. Reductive Photodegradation

Decomposition of PFOA takes place by two pathways during reductive processes, i.e., through breakage of C-C bond possibly without the formation of formate, and cleavage of carboxylic headgroup [64]. $e_{aq}^-$, one of the predominant species responsible for reductive photodegradation of PFAS can be generated under relatively milder experimental conditions under anaerobic conditions. $e_{aq}^-$ possess strong reducing capability ($E_{aq}/e^o = -2.9 \, V$) with strong affinity to halogenated organics. Fluorine, being the most electronegative (~4.0) among all atoms, possesses high ability to withdraw electrons and therefore could act as reductive reaction center for defluorination upon nucleophilic attack on PFAS (Bondel et al., 1989). According to the equation below, both oxidative ($^\bullet OH$) and reductive species including $e_{aq}^-$, hydrogen atoms ($^\bullet H$) could be generated upon water splitting under VUV irradiation [60].

$$H_2O + hv \, (< 190 \, nm) \; \rightarrow \; e_{aq}^-, \, {}^\bullet OH, \, {}^\bullet H, \, H_3O^+ \tag{7}$$

VUV at 185 nm could therefore degrade PFOA both by direct photolysis and by $e_{aq}^-$. Photolysis of iodide could also be used to generate $e_{aq}^-$. Iodide ion ($I^-$) upon UV absorption releases $e_{aq}^-$ via charge-transfer-to-solvent (CTTS) excitation of $I^-$ [66].

$$I^- + hv \; \rightarrow \; I_{CTTS}^{-*} \; \rightarrow \; I^\bullet + e_{aq}^- \tag{8}$$

$e_{aq}^-$ could also be generated in UV/sulphite process by photoionization of $SO_3^{2-}$. This process, however, requires a high pH and high concentration of S(IV) considering the low absorption and protonation of $SO_3^{2-}$.

$$SO_3^{2-} + hv \; \rightarrow \; SO_3^{\bullet-} + e_{aq}^- \tag{9}$$

All the above-mentioned processes have been investigated for reductive degradation of PFAS with varying degree of efficiency as is reviewed in Section 3.1. Other processes such as homogenous and heterogenous photocatalysis can also be used for the generation of different reductive species. The readers are directed to a review article focusing on these processes for the degradation of PFASs [15].

### 3.1. UV/VUV in Reductive Degradation of PFAS

Most of the studies looking at reductive degradation of PFAS have investigated the role of $SO_3^{2-}$ and $I^-$ in the presence of UV irradiation emitted from LP sources (Table 3). Some studies have also carried out comparative assessment of these two processes. One of such studies used UV irradiation (254 nm) and KI reporting a very high PFOA degradation (94%) and $F^-$ recovery (~77%) after 6 h; increasing the irradiation time increased the recovery of $F^-$ such that it reached ~99% after 14 h [67]. A comparative assessment with UV/$SO_3^{2-}$ showed the UV/$I^-$ process was markedly more effective considering only ~18% degradation of PFOA for UV/$SO_3^{2-}$. The reaction rate coefficient of degradation was $7.3 \times 10^{-3}$ min$^{-1}$ and $1.41 \times 10^{-3}$ min$^{-1}$ for UV/$I^-$ and UV/$SO_3^{2-}$, respectively. The degradation of PFOA present in wastewater from Teflon manufacturing plant was also tested [67]. The treatment was effective in degrading PFOA in real wastewater with 96% degradation after 12 h in the presence of 0.3 mM $I^-$ demonstrating the feasibility of the process for real wastewater. However, no details on the characteristics of water were provided and it is therefore difficult to conclude process efficiency based on their investigation [67]. It is therefore recommended to conduct future studies considering the water quality characteristics and other experimental conditions.

UV/$SO_3^{2-}$ process was also investigated in a study by Song et al. [68] for the degradation of PFOA using 254 nm UV irradiation and proposed mechanism of degradation (Figure 3). The authors also looked at the impact of reaction atmosphere. Under nitrogen atmosphere, the defluorination ratio of PFOA was 88.5% after 24 h whereas it was only 6.4% under oxygen atmosphere due to scavenging of $^\bullet H$ and $e_{aq}^-$. The process was severely impacted in the presence of $NO_2^-$ and $NO_3^-$ due to scavenging of $^\bullet H$ and $e_{aq}^-$. The main reducing species was confirmed to be $e_{aq}^-$ considering the comparable effect of $NO_2^-$ and $NO_3^-$ on PFOA degradation; the scavenging potential of $^\bullet H$ is almost 500-fold greater than $NO_3^-$ [69]. In agreement with the previous study by Qu et al. [67], the degradation was greater under alkaline conditions due to efficient generation of $e_{aq}^-$ at high pH [68]. Moreover, a 4-fold (0.5 to 20 mM) increase in the concentration of $SO_3^{2-}$ led to >12-fold increase in the defluorination ratio, i.e., from 5.6% to 68.6%. The relative quasi-stationary concentration of $e_{aq}^-$ and the degradation efficiency increased with increasing concentration of $SO_3^{2-}$ and pH demonstrating the role of $e_{aq}^-$ in reductive degradation of PFOA.

**Figure 3.** Proposed pathway of PFOA degradation during UV/$SO_3^{2-}$ reductive process under nitrogen atmosphere [68].

**Table 3.** Reductive degradation of PFAS compounds.

| Com pound | Concentrations (uM) | Matrix | pH | UV Source | Wave length | Oxidative/Reducing Agent | Degrdation (%) | Treatment Time (h) | Lamp Power (W) | Reference |
|---|---|---|---|---|---|---|---|---|---|---|
| PFOS | 32 | Ultrapure water | 9.2 | MP UV | 200–400 | $SO_3^{2-}$ | 98 | 0.5 | 250 | [70] |
| PFOA | 37.2 | milliQ, PBS | 7–11.2 | MP UV | - | $N_2O$, tert-butanol | | 6 | 500 | [56] |
| PFOA | 37.2 | DI, PBS, lake water, WWTP effluent | 4.3–9 | MP UV | - | none | | 6 | 500 | [65] |
| PFOA | 0.025 | $^{18}$O-water | 7 to 10 | LP UV | 254 | KI | 100 | 6 | 15 | [57] |
| PFOA | 24 | Ultrapure water | 4.5, 7.8, 10.3, 12 | LP UV | 185, 254 | none | | 3 | 23 | [64] |
| PFOA | 1.96, 2,51, 3.06 | Ultrapure water | | LP UV | 185, 254 | KI | 100 | 3 | 20, 110 | [71] |
| PFOA | 20 | DI | 10.3 | LP UV | 254 | $SO_3^{2-}$ | 100 | 1–24 | 10 | [68] |
| PFOA | 0.121, 1.21, 2.42 | Ultrapure water, tap water, river water | 3, 5.5, 7, 10 | LP UV | 185, 254 | none | | 3 | 20 | [23] |
| PFOS | 25 | milliQ | 9 | LP UV | 254 | KI | | 6 | 15 | [67] |
| PFDeA | 100 | DI | NG | LP UV | 185, 254 | $Na_2S$ | 100 | 6 | 23 | [40] |
| PFOA | 60.4 | aqueous solution | 3.7 | LP UV | 185, 254 | none | | 2 | 15 | [20] |

UV and VUV in the presence of KI was also employed by [71] to investigate the decomposition of PFOA using two different lamps having different intensities, i.e., UV intensities of 9.15 and 26.43 mW cm$^{-2}$ and VUV intensities of 5.05 and 20.73 mW cm$^{-2}$ for corresponding rate lamp power of 20 and 110 W. The reduction in PFOA concentration after 3 h irradiation differed markedly for 20 W UVC (31%) and 20 W VUV (87%) with 110 W VUV showing complete degradation. Incorporating KI to 20 W UVC led to an increase in degradation to 39% whereas a decrease was noted for 20 W VUV/KI (72%) indicating photo reductive degradation being less efficient than direct VUV photolysis. The trend was also apparent for TOC reductions with 20 W VUV/KI being most inefficient (~7%) and 110 W VUV being the most effective (~89%). Although the authors made a comparison with other studies such as Park et al. [72] and Qu et al. [67] with 2.6-fold greater and ~2-fold lower PFOA degradation, respectively, compared with 20 W UVC/KI, it is not possible to directly compare these findings. Firstly, the UVC/KI process is highly pH dependent and Qu et al. [67] found that alkaline pH (pH 9) highly favored the UVC/KI process whereas Giri et al. [71] used a much lower initial pH (5.5). Moreover, the findings of Qu et al. [67] were optimized in terms of the KI concentration. Furthermore, the UV lamps had different power output as well as time of treatment indicating large differences in the UV intensity making direct comparisons very difficult. A lower degradation of PFOA for 20 W VUV/KI compared with VUV alone was attributed to the photooxidation rate value of $64.2 \times 10^{-2}$ h$^{-1}$ which was 1.5-fold greater than that for $e_{aq}^{-}$. However, it should be emphasized that these findings were based on a single experiment without taking into consideration the impact of KI and pH on the process efficiency. Defluorination ratio varied between different processes but followed trends similar to degradation of PFOA, i.e., the ratios were 0.5 and 3.5, 20.5 and 7.7, and 69% for 20 W UVC and UVC/KI, 20 W VUV and VUV/KI, and 110 VUV, respectively, after 3 h.

As mentioned above, Qu et al. [67] optimized PFOA degradation in terms of the concentration of iodide and noted that the degradation increased with increasing concentration from 0.1 to 0.3 mM before decreasing with increased concentration of I$^{-}$. Briefly, it was attributed to increasing concentration of triiodide to a level that it started to scavenge $e_{aq}^{-}$. The mechanism of degradation was found to be reductive cleavage of C-F bonds by $e_{aq}^{-}$ and

simultaneous destruction of C-C bonds by UV irradiation. In their later work, Qu at al. [57] showed that the reductive degradation of PFOA using UV/KI system was highly pH dependent. They investigated pH range of 5–10 and found that the rate constant was 49-fold greater at pH 10 than at pH 5. The defluorination efficiency was only 5.9% after 6 h at pH 5 whereas it increased to 44.7% at pH 8 and 69.8% at pH 10. The effect was attributed to two main factors. Firstly, in the presence of $H^+$ or low pH, $e_{aq}^-$ were converted to $H^\bullet$ at rate constant of $2.3 \times 10^{10}$ $M^{-1}s^{-1}$ [29]. Secondly, it has been shown that at high pH values, specifically >8.5, $I_2$ could disproportionate into $I^-$ and $IO_3^-$ [73] that led to more iodide ions being recycled, lower generation of triiodide and iodine, and hence greater generation of $e_{aq}^-$. At pH values higher than 7, $e_{aq}^-$ was the main species leading to reduction of PFCA through cleavage of C-C and C-F bonds. A recent study [27] also suggested a shift from oxidizing to reducing environment with an increase in pH during $UV/VUV/I^-$ process.

Some studies have looked at the impact of water matrix and other experimental conditions (pH, initial concentration of PFAS) on degradation kinetics. Song et al. [68] modelled PFOA degradation in ultrapure water (UPW), surface water (SW) and wastewater (WW) during $UV/SO_3^{2-}$ [68]. No difference in the degradation was observed for UPW and SW whereas it was much slower in WW which could be due to higher concentration of $Cl^-$ in WW (1.51 mM) compared with SW (0.12 mM) since the effect of $NO_3^-$, $PO_4^{3-}$ and alkalinity was found to be negligible. It must be noted that the WW also had a high TOC concentration (~17.7 mg $L^{-1}$) compared with SW (0.198 mg $L^{-1}$). It has earlier been reported that NOM could activate PS [74] and generate $SO_4^{\bullet-}$. In this study, however, the role of NOM was not elucidated in detail and it is therefore not possible to relate their findings with th study of Fang et al. [74]. It is recognized that the radical chemistry in the presence of NOM is complex and needs further investigation. Nonetheless, the model developed for the simulation of pH changes and the impact of carbonate and chloride scavenging on PFOA degradation during UV/PS process fitted well with the experimental data.

Giri et al. [23] also investigated reductive degradation of PFOA in different water matrices. PFOA degradation in ultrapure water, tap water and river water was 87%, 57%, and 37%, respectively, after 3 h of VUV treatment [23]. Similarly, defluorination ratio was almost half for tap water than for UP water whereas that for river water was almost 7-fold lower. The trend observed was therefore more representative of the conventional UV/VUV-based processes since the concentration of alkalinity and non-purgeable organic carbon followed the trend of PFOA degradation. However, the concentration of other constituents such as ammonia which is known to negatively impact PFOA degradation was not investigated. These results differ from Lyu et al. [65] who reported negative impact of lower ionic strength; Giri et al. [23] showed greater degradation of PFOA for UPW despite it having the lowest ionic strength among the water matrices tested. It is however possible that the impact of ionic strength was overshadowed by the presence of other organics.

A systematic study of the effects of pH on reductive degradation of PFOA by VUV was investigated in nitrogen and oxygen atmosphere [64]. Total degradation of PFOA (initial concentration of 24 μM/L (10 mg $L^{-1}$) at original pH of 4.5 both in nitrogen and in oxygen atmosphere was observed after 3 h. An increased formation of fluoride ions was noted with increased irradiation time leading to comparable defluorination ratio under studied conditions, i.e., 50.6% and 49% under oxygen and nitrogen atmosphere, respectively. However, at pH 12, the reaction was much faster in nitrogen atmosphere than in oxygen due to generation of $e_{aq}^-$ and degradation of PFOA [64]. The degradation of PFOA increased with increasing pH in nitrogen atmosphere with the largest increase occurring when pH increased from 10.3 to 12. Likewise, defluorination ratio increased demonstrating the by-products were decomposed faster under alkaline conditions. An opposite trend was found in oxygen atmosphere and the degradation of PFOA and defluorination decreased with increasing pH indicating the importance of pH in the VUV process. Similar findings were reported by Giri et al. [23] where presence of dissolved oxygen was found to inhibit the degradation of PFOA. However, the trend in the degradation of PFOA with increase in

pH was different to that observed by Wang and Zhang [64]. Giri reported a small increase in PFOA degradation with decreasing pH from initial value of 5.5 (88%) to 3 (87.3%) whereas increasing the pH to 7 decreased the degradation to 77.9%. A further increase in pH to 10 increased the degradation to 83% which agrees with the findings of Wang and Zhang [64] who reported an increase in degradation with increasing pH, as mentioned above.

The difference in the degradation of PFOA could be considered marginal as it was within ~10% for all the pH values. Under alkaline conditions, the absorption of VUV is known to be very high for hydroxide ion ($OH^-$) considering its high coefficient of absorption (3000 $mol^{-1} \cdot cm^{-1}$) [75] resulting in reduced absorption by PFOA and hence reduced degradation. It is also important to note that $e_{aq}^-$ could be scavenged by oxygen leading to reduction in PFOA degradation. In the presence of nitrogen, the yield of $e_{aq}^-$ as well as their lifespan was shown to increase due to absence of oxygen leading to enhanced degradation of PFOA. An increase in PFOA degradation with increasing pH as observed by Wang and Zhang [64] could be attributed to $e_{aq}^-$ but similar was not apparent in the study of Giri et al. [23]. These two different findings warrant further investigation into the role of acidic pH and $e_{aq}^-$ for the degradation of PFOA in the presence of nitrogen gas.

It is worth noting that both these studies used fairly similar UV lamps in terms of lamp power (23 W, 20 W) and time of irradiation (3 h), however the initial concentration of PFOA was markedly different, i.e., 24 (10 mg $L^{-1}$) cf. 2.42 (1 mg $L^{-1}$), for Wang and Zhang [64] and Giri et al. [23], respectively. Although Giri et al. [23] investigated the impact of PFOA concentration on total degradation, the concentration range used was narrow (0.05, 0.5 and 1 mg $L^{-1}$). Despite insignificant impact on total degradation of PFOA, defluorination ratio was higher for lower concentration of PFOA such that it increased by 1.3- and 3.5-fold for 2- and 20-fold decrease in PFOA concentration, respectively. This trend could be attributed to reduced photon energy per unit PFOA concentration leading to slower removal at high concentrations which could also minimize the mineralization of short-chain by-products [23].

Reductive degradation is an efficient method for the degradation of PFAS. Intensification of reductive degradation processes require novel approaches to improve their potential for practical applications. For example, the process could be intensified in the presence of compounds containing carboxylate groups that are bound to nitrogen atoms such as Aminopoly(carboxylic acid)s. Hence, using photosensitizers such as nitrilotriacetic acid could facilitate enhanced degradation of PFASs since it scavenges $^\bullet OH$ leading to their reduced recombination with $e_{aq}^-$ [76]. An increasing number of studies have reported the impact of various experimental parameters on reductive degradation of PFAS. However, the impact of water quality on reductive degradation of PFAS needs to be investigated since it might affect the degradation process differently than known degradation trends for conventional organics during UV-based processes.

### 3.2. Medium Pressure UV in Reductive Degradation of PFAS

Using a medium pressure lamp (500 W), Lyu et al. [56] investigated the degradation of PFOS in the presence of selective promotors or inhibitors including $O_2$, $H_2O_2$, $N_2O$, tert-butanol and by adjusting the pH (2.6–11.8) and temperature (35–100 °C). The authors found that the degradation of PFOS primarily occurred via reduction using hydrogen atoms and/or $e_{aq}^-$ and it enhanced with an increase in temperature and pH. These findings agree with those of Wang and Zhang, [64] who also reported an increase in the degradation of PFOA with increasing pH due to higher generation of $e_{aq}^-$ during 185 nm VUV process. Furthermore, the increase in degradation was noted with an increase in temperature from 35 to 100 °C. Similar to the findings of Wang and Zhang, [64] the degradation was suppressed under oxygen environment whereas it increased substantially in the presence of tert-butanol. Degradation was also suppressed markedly in the presence of $N_2O$ that could be attributed to the potential conversion of $e_{aq}^-$ and hydrogen atoms to $^\bullet OH$ which are unable to degrade PFOA under mild conditions as reported by others [29,77]. It was therefore concluded that the degradation of PFOS took place via photo-reductive path since

direct photolysis was not possible to occur in the system considering low $pK_a$ value of PFOS ($-3.27$) and suppression of degradation in the presence of oxidative agents. Maximum degradation rate ($0.91$ $h^{-1}$) was reported under high pH ($11.8$) and temperature ($100\,^\circ C$) without using any chemical. It however must be noted that these conditions are not feasible for practical applications. It is therefore important to develop methods/processes that could more efficiently generate reductive species such as $e_{aq}^-$ under viable conditions to facilitate practical applications. For example, it was shown that the species like $e_{aq}^-$ could be generated by flash photolysis of solutions containing inorganic salts and aromatic compounds including amino acids [78].

Ability of $e_{aq}^-$ to vigorously attack non-target species leads to competitive reactions and therefore reduced efficiency in degrading target contaminants. The efficiency of $e_{aq}^-$ in degrading PFOA could be improved by high photon flux ($9.9 \times 10^{-8}$ Einstein/cm$^2$·s), for example, using a high-pressure mercury lamp ($200$–$400$ nm) in a $UV/SO_2^{3-}$ system [70]. The decomposition of PFOS (initial concentration $32$ μM) was $98\%$ after $30$ min at pH $9.2$. Decomposition kinetics were described as pseudo-first-order with reaction rate constant of $7.08$ $h^{-1}$. A preliminary comparison of pseudo-first-order rate constants with other reductive process such as UV/alkaline 2-propanol ($0.039$ $h^{-1}$) [43], and UV/KI ($0.18$ $h^{-1}$) [72] demonstrated that $UV/SO_2^{3-}$ system was much more effective than other reductive processes. The authors also investigated the $UV/S_2O_8^{2-}$ and UV/KI processes and found no noticeable degradation of PFOA after $30$ min [70]. Since $NO_3^-$ is a strong scavenger of $e_{aq}^-$, the authors investigated its role at concentrations of $0.5$, $1$ and $2$ mM $NO_3^-$. At the lowest concentration of $0.5$ mM $NO_3^-$ ($1.3$ mg N L$^{-1}$), the process resulted in ~$94\%$ degradation of PFOS with total degradation of $NO_3^-$ during first $15$ min of $UV/SO_2^{3-}$ process. Increasing the concentration of $NO_3^-$ to $1$ and $2$ mM, however, severely impacted the decomposition of PFOS with total suppression at the highest concentration tested. In another study, monochloroacetic acid (MCAA) which is used as the model compound to investigate the reductive efficiency of a system, reported the inhibition of MCAA dechlorination by $e_{aq}^-$ at a much lower concentration of $NO_3^-$, i.e., $1.3$ mg N/L [69]. These findings demonstrate the impact of high photon flux system in overcoming some of the drawbacks related to scavenging of $e_{aq}^-$ by $NO_3^-$. The process could also be considered more environmentally friendly considering $SO_4^{2-}$ as a more benign reaction product compared with KI and ferrocyanide [70].

The impact of pH on degradation was significant with greater decomposition of PFOS at pH > $9.2$ and almost no degradation at pH $7$ [70]. It is worth noting that $SO_3^{2-}$ exhibits strongest UV adsorption among sulphite species in alkaline conditions [70]. Moreover, the generation of $e_{aq}^-$ is strongly dependent on the concentration of $SO_3^{2-}$ in $UV/SO_2^{3-}$ process [68] indicating $e_{aq}^-$ as the predominant reducing species in the studied system. It was previously mentioned that quenching of $e_{aq}^-$ occurs at low pH by $H^+$ that generates •H [57]. Overall, it can be concluded that pH above $8$ is favorable for the generation of $e_{aq}^-$ leading to faster degradation kinetics of PFOS during $UV/SO_2^{3-}$ process. The concentration of short chain PFCA (PFHpA, PFHxA, PFPeA and PFBA) was maximum after $10$–$15$ min followed by a decrease with increasing time of reaction. The concentration of fluorine kept increasing up to $60$ min indicating that defluorination continued after complete PFOS decomposition yielding a final defluorination efficiency of ~$70\%$. Considering the total PFOS decomposition and high defluorination efficiency, it can be concluded that the impact of competitive reactions could be minimized by promoting the generation of $e_{aq}^-$ through high photon flux system employing $UV/SO_2^{3-}$. These findings show tremendous potential of practical applications but the impact of other water constituents such as organic matter and dissolved oxygen needs to be investigated.

A potential treatment for achieving greater reduction of PFASs is combining of $SO_2^{3-}$ and $I^-$ with UV irradiation. The process has been proven to produce synergistic effect for the degradation of monochloroacetic acid (MCAA) due to the continuous generation of $e_{aq}^-$ that was attributed to the cycling of $I^-$ in the presence of S(IV) [69]. Since efficient

generation of $e_{aq}^-$ is the key to reductive processes, this process needs to be investigated for its application in the degradation of PFAS. Work on the toxicity and biodegradability of intermediate by-products also needs to be investigated.

### 3.3. Reaction By-Products during Reductive Degradation

Investigating the degradation of PFOA during VUV process, Wang and Zhang [64] identified six shorter-chain PFCAs, i.e., PFHpA, PFHxA, PFPeA (perfluoropentanoic acid), PFBA (perfluorobutanoic acid), PFPrA and TFA (trifluoroacetic acid) irrespective of the oxygen or nitrogen atmosphere. Similar by-products were reported during VUV/$Fe^{3+}$ by [23]. Evolution and concentration of these intermediates was, however, different in two reaction atmospheres with their concentration generally lower in nitrogen atmosphere than in the presence of oxygen. Additionally, the authors identified formate ion demonstrating the cleavage of the headgroup of carboxylate of PFOA and short-chain PFCA molecules. The decomposition of PFOA during oxidative processes generally initiates with decarboxylation leading to the formation of formate and PFHpA [28,79]. During VUV photolysis of PFOA, Chen et al. [20] identified four by-products including PFHpA, PFHxA, PFPeA and PFBA which are similar to those identified by Wang and Zhang [64]. In both these investigations, the PFHpA appeared first and reached its maximum concentration in the first h of treatment before decreasing. This was expected considering it contains seven carbon atoms; the intermediates with less carbon atoms appeared later. Overall, the trend in both studies showed that the intermediates having longer chains were formed initially before breaking down into shorter chains in a stepwise manner.

Giri et al. [71] looked at the concentration profiles of PFOA in addition to intermediates formed during UV and VUV in the presence of KI. VUV (110 W) yielded greater concentration of six intermediates (PFHpA, PFHxA, PFPeA, PFBA, PFPrA, TFA) whereas only four intermediates (PFHpA, PFHxA, PFPeA, PFBA) were observed for 20 W VUV/KI at much lower concentrations. The similar intermediates were observed for 20 W VUV but at much higher concentrations than for 20 W VUV/KI. However, PFHpA and PFHxA were the only short-chain intermediates observed for 20 W VUV with and without KI. Only PFHpA and PFHxA were observed during 20 W UVC alone and in combination with KI. Using UVC/KI, formation of intermediates during PFOA degradation was also investigated by Qu et al. [67]. In addition to fluoride ions, the intermediates formed included formic acid, acetic acid, and six short-chain PFCAs namely PFHpA, PFHxA, PFPeA, PFBA, PFPrA and TFA [67]. It is worth noting that the similar intermediate by-products were reported by Giri et al. [71] during VUV treatment as mentioned above. The concentration of longer chain intermediates (PFHpA, PFHxA) reached its maximum after 1 h irradiation whereas those with $C_1$–$C_3$ kept increasing up to 2 h demonstrated their degradation as the process continued [67].

In their later study, Qu et al. [57] investigated the formation and decomposition of short-chain intermediates at different pH values. At pH 5, the concentration of PFHpA increased with increasing reaction time such that it reached 3.9 µmol $L^{-1}$. Hydrogen atoms were the main species responsible for the breakage of C-C bonds with little impact on the C-F bond of PFOA. Lower pH led to accumulation of intermediates which was due to reduced concentration of $e_{aq}^-$. An increase in pH reduced the time needed to reach the maximum concentration of PFHpA, i.e., it reached maximum level after 1 h when pH was ≥7. Furthermore, the concentration was much lower for higher pH values (7–10) indicating that higher pH led to reduced accumulation of degradation intermediates (PFHpA, PFHxA, PFPeA, PFBA, PFPrA, TFA) [57]. For example, the concentration of PFHpA at pH 7 was 2.5-fold greater (1.27 µmol $L^{-1}$) than at pH 10. Since the predominant species for reductive degradation were $e_{aq}^-$, their abundance at higher pH values ensured efficient breakage of C-F and C-C bonds resulting in lower accumulation of intermediates.

Song et al. [68] identified degradation intermediates using UV/$SO_3^{2-}$ process for the degradation of PFOA using a 254 nm UV lamp. The reaction intermediates identified were PFCAs with 2–7 carbon atoms whose concentration increased during first ~30–90 min

followed by a decrease demonstrating stepwise degradation of both PFOA and its intermediates. In addition to PFCA reaction intermediates, two other groups of fluorine containing compounds were identified. The first group comprised of $C_7F_{14}HCOOH$, $C_7F_{13}H_2COOH$, $C_6F_{12}HCOOH$, $C_5F_{10}HCOOH$, $C_4F_8HCOOH$ and $CF_2HCOOH$ indicating the reductive degradation of PFOA as indicated by the cleavage of C-F bond by $e_{aq}^-$. The second group consisted of fluorinated alkyl sulfonates ($C_7F_{15}SO_3^-$, $C_6F_{13}SO_3^-$, $C_5F_{11}SO_3^-$, $C_4F_9SO_3^-$, and $C_3F_7SO_3^-$). It was, however, reported that the concentration of F-containing intermediates was much lower (0.02 $\mu$mol $L^{-1}$) than the initial concentration of PFOA (20 $\mu$mol $L^{-1}$).

## 4. Future Work and Research Needs

Despite significant research being carried out on the degradation of PFAS, there are several areas requiring significant work before adopting technological solutions. Firstly, there is a need to investigate the impact of the presence of organics and co-contaminants on the degradation of entire suite of PFAS compounds. Most studies have either used simple water matrix and those who have investigated the impact of water quality characteristics has mainly focused on the impact of single co-contaminant and are therefore not representative of a real water matrix. There is also a need to reduce the cost of treatment, for example, through multiple and complementary treatments to minimize the duration of energy intensive UV-based processes. The focus is therefore needed to be on the partial degradation of most recalcitrant part of PFAS making them more amenable to removal by simple and more economical pre- and post-UV based treatments such as biological processes that could stimulate specific bacteria to selectively degrade remaining PFAS and/or their intermediates. Both oxidative and reductive processes are promising but further research is needed to establish their efficiency for targeted PFAS degradation in combined treatment approaches under more challenging conditions. The outcomes from such studies would contribute to assessment of the feasibility of these processes at large-scale. None of these studies examined the toxicity of water after UV treatment, which should be focused in future studies to understand how toxicity changes during different treatments and to ensure safety of treated water.

Although $^\bullet$OH are unable to directly oxidize PFAS, they have been shown to weaken the C-F bond in the presence of other substrates including heat and UV. Since $^\bullet$OH based processes are well established with some of the processes having applications at full-scale, it is logical to investigate these processes from the viewpoint of enhancing their efficiency for this particular class of compounds. This requires understanding of the reaction mechanisms in the presence of potential substrates and/or combination of conditions that could promote breakage of the C-F bond. For example, a combination of low pH (pH $\leq$ 3.5) in the presence of $^\bullet$OH during UV/$Fe^{3+}$ process was found to be favorable for the regeneration of $Fe^{3+}$ and enhanced defluorination efficiency. Similarly, the degradation of PFOA attributed to $^\bullet$OH occurred only in the presence of 4-methoxyphenol, a phenolic co-substrate [80]. This effect was attributed to intermediate radical species that were generated leading to non-specific degradation of PFOA through damaging and/or weakening the highly stable C-F bond.

It has been reported that the presence of $Cl^-$ could interfere the complexation between $Fe^{3+}$ and PFOA depending on the source of iron used [24]; the decomposition of PFPeA was lower when FeCl3·6H2O was used compared with $Fe_2(SO_4)_3$·7.5H$_2$O. However, in another study [39], defluorination ratio was unaffected by the presence of $Cl^-$ and the effect of $SO_4^{2-}$ was considered more important particularly at higher concentrations. In fact, it has been proposed that the $Cl^\bullet$ generated by reaction between $Fe^{3+}$ and $Cl^-$ in presence of UV [81] might also contribute to decomposition of PFOA [82] highlighting the need to identify the conditions (concentration and type of co-existing ions) that affect the degradation of PFOA during UV-based process. Similarly, the role of $SO_4^{2-}$ needs to be investigated further in UV/$Fe^{3+}$ process since it affects the complexation of $Fe^{3+}$ and PFOA at higher concentrations [39].

Although UV photo-oxidative and reductive processes are able to degrade PFAS, given the diversity and unique properties of PFAS properties, these processes must be viewed

from a different perspective. It could involve developing treatment trains considering concentration of different PFAS compounds relative to other water quality parameters including co-exiting ions, complexing agents, organic matter and other contaminants. Approaches employing sequential processes including reductive degradation to generate less fluorinated compounds that could be oxidized and mineralized by oxidative and biological processes, respectively, could be needed to achieve enhanced PFAS degradation. Further research on the optimization and cost of such treatment trains needs to be investigated for designing large-scale processes. Biological processes could potentially play a very important role in reducing the cost of combined treatments through careful optimization of oxidative/reductive processes and by stimulating growth of selective bacteria for the degradation of reaction by-products.

## 5. Conclusions

Chemical reductive processes appear to be more promising than oxidative ones with PFOA as the most widely investigated compound class by UV driven processes. Oxidative processes, however, are more promising for larger-scale applications in general and are easier to be integrated into existing approaches. Oxidative processes have mostly employed single emission low pressure UV lamps emitting at 254 nm whereas reductive processes have mainly employed VUV (emitting at both 254 and 185 nm) for degradation of PFAS. The oxidative processes need much longer irradiation treatment time for achieving PFAS degradation compared with reductive processes. Overall, the findings of the laboratory studies are diverse in terms of the time of irradiation, water quality and type and initial concentration of PFAS. Most studies have elaborated prevailing mechanism of degradation with comparable findings in terms of their types and evolution during both oxidative and reductive processes. However, most studies looked at their formation using simple water matrix and therefore more research is needed under different water quality conditions to investigate how degradation pathways and consequently by-product formation change in the presence of co-contaminants.

Since PFAS are difficult to be degraded using a single treatment step, it is important to develop and investigate processes aimed at targeted degradation of PFAS. Such approaches must therefore consider the impact and removal of co-contaminants and ions considering the complexity of direct and indirect reactions between PFAS and oxidizing and reducing species. One of the challenges all micropollutants pose is their relative lower concentration compared with other contaminants in real water matrix. PFAS, however, makes it even more challenging due to their persistent nature requiring modifications in existing treatment approaches that could be based on selective and prior removal of certain constituents (for example, $Cl^-$) to minimize issues with complexation of PFAS with an oxidant (i.e., $SO_4^{\bullet-}$). Other approaches could include facilitating regeneration of oxidants (i.e., $Fe^{3+}$) that could occur in the presence of other species ($^{\bullet}OH$) for enhanced degradation of PFAS.

**Funding:** This work was funded by NIVA's publication funding initiative.

**Institutional Review Board Statement:** Not applicable.

**Informed Consent Statement:** Not applicable.

**Data Availability Statement:** Not applicable.

**Conflicts of Interest:** The author declares no conflict of interest.

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
