# Peer review of "Reductive and Oxidative UV Degradation of PFAS—Status, Needs and Future Perspectives"

_water, doi:10.3390/w13223185_

Round 1

Reviewer 1 Report

The manuscript by M. Umar reviews the fundamental mechanisms and recent development of UV-based oxidative and reductive processes for PFAS degradation. The referencing is not enough. In my opinion, the paper can be accepted to the Journal after major revisions (see below).

Specific comments:

(1) The overall problem of this review is that it emphasizes too detailed discussions and direct quotations of literature data, ignoring digestion and refining important conclusions. My personal suggestion is to delete some overly detailed discussion and add a short summary at the end of each chapter and section.

(2) This review only lists two tables. In fact, the author can consider using schematic diagrams in the literature to explain the mechanism and pictures/figures to explain some important data. The main purpose is to help readers obtain information at a glance.

(3) The reference format of this manuscript must be corrected in order to meet the requirements of the Journal.

(4) Line 62, “While several review papers have been published reporting different approaches for the removal and degradation of PFASs, …” The author should add a list of these review paper for readers.

(5) Section 2.1, the author should add one or two sentences to give some general information on the UVC and VUV.

(6) Line 126, “UV-based AOPS”, the author should provide the full name of the technology AOP, followed by its abbreviation.

(7) Some sentences need to be slightly modified or added with commas so that the reader can easily understand their meaning. For example, Line 892, “None of the studies have looked into toxicity of water after UV-based treatments which should be focused in future studies to understand how toxicity changes during different treatments and to ensure safety of treated water.” It can be modified to “None of these studies examined the toxicity of water after UV treatment, which should be focused in future studies to understand how toxicity changes during different treatments and to ensure safety of treated water.” Line 919, “Although UV photo-oxidative and reductive processes are able to degrade PFAS, there is a need to look at these processes from a different perspective considering the diversity and unique characteristics of PFAS properties.” It could be modified to “Although UV photo-oxidative and reductive processes are able to degrade PFAS, given the diversity and unique properties of PFAS properties, these processes must be viewed from a different perspective.”

(8) This review seems a bit outdated. The author failed to cite the literature of the last three years, especially in recent UV-based advanced oxidation processes (AOP) for degradation of PFOA and PFOS. The author needs to perform a literature search again to include the latest literature.

A review on degradation of perfluorinated compounds based on ultraviolet advanced oxidation, Environmental Pollution, Volume 291, 15 December 2021, 118014, https://doi.org/10.1016/j.envpol.2021.118014

Removal of polyfluorinated telomer alcohol by Advanced Oxidation Processes (AOPs) in different water matrices and evaluation of degradation mechanisms, Journal of Water Process Engineering, Volume 39, February 2021, 101745, https://doi.org/10.1016/j.jwpe.2020.101745

Photochemical decomposition of perfluorochemicals in contaminated water, Water Research, Volume 186, 1 November 2020, 116311, https://doi.org/10.1016/j.watres.2020.116311

Yang, Y., Zhang, Q., Chen, B. et al. Toward better understanding vacuum ultraviolet—iodide induced photolysis via hydrogen peroxide formation, iodine species change, and difluoroacetic acid degradation. Front. Environ. Sci. Eng. 16, 55 (2022). https://doi.org/10.1007/s11783-021-1489-0

Role of oxygen and superoxide radicals in promoting H2O2 production during VUV/UV radiation of water, Chemical Engineering Science, Volume 241, 21 September 2021, 116683

https://doi.org/10.1016/j.ces.2021.116683

Removal of Chlorinated Organic Pollutants from Groundwater Using a Vacuum-UV-Based Advanced Oxidation Process, Debra Barki, Sara Sabach, and Yael Dubowski, ACS EST Water 2021, 1, 9, 2076–2086

https://doi.org/10.1021/acsestwater.1c00167

Highly efficient degradation of perfluorooctanoic acid: An integrated photo-electrocatalytic ozonation and mechanism study, Chemical Engineering Journal, Volume 391, 1 July 2020, 123533, https://doi.org/10.1016/j.cej.2019.123533

Reviewer 2 Report

The review is interesting and well-constructed, but it is flawed by the lack of information of the role of ozone generation in the PFAS degradation. In fact, the wavelength of 185 nm is highly efficient in generating ozone, but it is never cited in the manuscript as well as is role in the PFAS oxidation. A chapter about the ozone role is required.

Round 2

Reviewer 1 Report

The manuscript can be accepted in the current form.

Reviewer 2 Report

The authors corrections have improved the manuscript. I recommends its publication.